

# Molecular characterization of organic aerosol in Himalayas: insight from
# ultra-high resolution mass spectrometry
Yanqing An[1], Jianzhong Xu[1], Lin Feng[1, 2], Xinghua Zhang[1, 2], Yanmei Liu[1, 2], Shichang Kang[1,2]
[1]State Key Laboratory of Cryospheric Science, Northwest Institute of Eco-Environment and
Resources, Chinese Academy of Sciences, Lanzhou 730000, China
[2]University of Chinese Academy of Sciences (UCAS), Beijing 100049, China
Corresponding Author: Jianzhong Xu, jzxu@lzb.ac.cn



**Abstract**
An increasing trend in aerosol concentration has been observed in Himalayas in recent years, but
the understanding of the chemical composition and sources of aerosol remains poor. In this
study, molecular chemical composition of water soluble organic matter (WSOM) from two filter
samples (denoted as F30 and F43) collected during high aerosol loading periods at a high altitude
station (Qomolangma Station, QOMS, 4276 m a.s.l.) in the northern Himalayas were identified
by positive electrospray ionization Fourier transform ion cyclotron resonance mass spectrometry
(ESI-FTICR-MS). More than 4500 molecular formulas were identified in each filter sample
which were classified into two compound groups (CHO and CHON) based on their elemental
composition with both accounting for nearly equal contributions in number (45% – 55%). The
relative abundance weighted mole ratio of $O/C_w$ for F30 and F43 are 0.43 and 0.38, respectively,
and the weighted double bond equivalent ($DBE_w$), an index for the saturation of organic
molecules, were 6.26 and 6.92, respectively, suggesting their medium oxidation and saturation
degrees. Although the $O/C_w$ mole ratio was comparable for CHO and CHON compounds, the
$DBE_w$ was significant higher in CHON compounds than CHO compounds. More than 50%
molecular formulas in Van Krevelen (VK) diagram (H/C vs. O/C) located in 1 – 1.5 (H/C) and
0.2 – 0.6 (O/C) regions, suggesting potential lignin-like compounds. The distributions of CHO
and CHON compounds in VK diagram, DBE vs. number of C atoms, and other diagnose
diagrams showed highly similarities between each other suggesting their similar source and/or
atmospheric processes. Detailed molecular information in the common formula of these two
filters was explored. Many formulas with their homologous series of compounds formed from
biogenic volatile organic compounds and biomass mass burning emitted compounds were found
in the WSOM with high relative abundance suggesting the important contribution of these two
sources in Himalayas. The high DBE and high nitrogen containing of aerosol would have
important implication for aerosol light absorption and biogeochemical cycle in this remote
region.

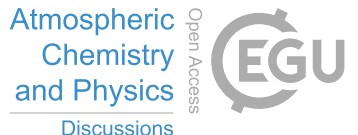

## 1. Introduction

Elevated pollutant concentrations has been frequently observed over Himalayas during pre-monsoon period (March to June) (Bonasoni et al., 2010). The high aerosol loading plume are originated from the southern regions of Himalayas such as northwestern India and/or Indian Gangetic region based on air mass back trajectory analysis and satellite observation (Liu et al., 2008; Lu et al., 2012; Lüthi et al., 2015). In recent decades, due to increased consumption on fuels (including biofuels and fossil fuels) by industry and residents, air pollution has been a serious issue in South Asia (Gustafsson et al., 2009). Accompany with favorable atmospheric circulation, air pollutants emitted or formed in these regions can be fast transported to Himalayas and Tibetan Plateau (HTP) (Xia et al., 2011).

Elevated aerosol concentration for the pristine region of the HTP is thought to have essential climate and environment effects. For example, the transported aerosol could heat the air at the higher layer of troposphere over the HTP and impact on the monsoon system of south Asia and accelerate the melting of glacier in Himalayas (Lau et al., 2006; Ramanathan et al., 2007). This heating effect is predominantly from the light absorbing particular aerosol (LAPA) such as black carbon (BC) and brown carbon which is part of organic aerosol (OA) (Ram et al., 2010; Zhang et al., 2015; Zhang et al., 2017). BC come from incomplete combustion and dominates the absorption of LAPA; Brown carbon could from many processes such as primary emission and secondary process and have an increasing contribution (up to ~20%) to the light absorption in recent years (Laskin et al., 2015). Many studies show that open biomass burning is an important source of BC and brown carbon (e. g., Saleh et al., 2014), which is very popular in developing regions in the southern of Himalayas. However, high elevation and mixed biomass fuels in these regions could make the evolution of biomass burning emission more complicated and the chemical information of OA remains poorly characterized so far (Fleming et al., 2018).

The details on the molecular composition of OA are important for understanding the sources and chemical evolution of OA (Laskin et al., 2018). Previous studies conducted in the HTP have focused on a limited number of molecular markers such as organic acids which are closely related with biomass burning emission (Cong et al., 2015), and some toxicology species such as



polycyclic aromatic hydrocarbons (PAHs) and persistent organic pollutants (POPs) which are
related with anthropogenic activities (Wang et al., 2015; Wang et al., 2016). In addition, online
measurement using Aerodyne high resolution time-of-flight aerosol mass spectrometer (HR-
ToF-AMS) had provided more details on the OA chemistry and sources with high time
resolution (Xu et al., 2018). However, different instrument has its limitations on OA detection
and ultra-high mass resolution of mass spectrometry which can identify a large number of
molecular formulas is lacking.

Electrospray ionization (ESI) with ultrahigh-resolution Fourier transform-ion cyclotron
resonance mass spectrometry (FTICR-MS) can be used to identify the individual molecular
formula of complex mixture because of its extremely high resolution and mass accuracy
(Mazzoleni et al., 2010). In this study, we focus on the comprehensive characterization of the
molecule composition of water soluble organic compound in fine particle aerosol collected in the
northern slope of central Himalayas using positive mode ESI-FTICR-MS.

## 2. Methodology
### 2.1. Aerosol sampling
Field study was conducted at the Qomolangma Station (QOMS, 28.36° N, 86.95° E, 4276 m
a.s.l.) located at the toe of Mt. Qomolangma from Apr. 12 to May 12, 2016 using a suit of
instruments (Zhang et al., 2018b), and the instruments used in this study includes a HR-ToF-
AMS (Aerodyne Research Inc., Billerica, MA, USA) for 5-min size-resolved chemical
compositions (organics, sulfate, nitrate, ammonium, and chloride) of non-refractory submicron
particulate matter (NR-PM$_1$) and a photoacoustic extinctionmeter (PAX, DMT Inc., Boulder,
CO, USA) for BC mass concentration. The QOMS observatory locates at a remote site with
sparse local residents and anthropogenic activities around it, except for a high way for the
tourism to the west about 500 m. The tourists are normally increased from June each year due to
the warmer weather during summer. The weather at the QOMS during the field study was
dominated by westerlies with the prevailed wind from west and southwest with an average air
temperature of 5.7 ± 5.0 °C. A low-volume (16.7 L min$^{-1}$) particular matter (PM) sampler (BGI,
USA, model PQ 200) with an aerodyne diameter cutoff of 2.5 μm at the inlet was used to collect



PM$_{2.5}$ filter samples on pre-baked quartz fiber filters (47 mm, Pall Life Science, NY, USA). Due
to the low aerosol loading at this remote region, two days sampling strategy was adapted for each
filter collection starting from 8:00 am to 7:45 am at the day after tomorrow (local time). A total
of 18 filter samples were collected during the field study with three procedure blanks which were
used to assess potential contamination during sampling and transportation. The sampling air
volume ranged from 35.1 to 48.1 m$^3$ at ambient conditions.  Two filter samples collected during
Apr. 25 – 27 (F30) and Apr. 29 – May 1 (F43) were chosen in this study due to the relative
higher aerosol loading based on HR-ToF-AMS results (section 3.1) and distinct particulate
matter on the filter. In addition, the temporal variations of aerosol concentration recorded by HR-
ToF-AMS during these two filter periods shown a smoothly increase indicating a regional
transportation which could be typical long-range transport events at this region.

**2.2. Chemical analysis**
For FTICR-MS analysis, these two filters were firstly extracted in 20 mL Milli-Q water in an
ultrasonic bath for 30 min and filtered using 0.45 µm pore-size Acrodisc syringe filters to
remove water insoluble matter (Pall Science, USA). Prior to FTICR-MS analysis, the extraction
was concentrated and purification using PPL (Agilent Bond Elut-PPL cartridges, 500 mg, 6 mL)
solid phase extraction (SPE) cartridges for water soluble organic matter (WSOM). Note that
through SPE cartridge, the most hydrophilic compounds such as inorganic ions, and low-
molecular-weight organic molecules such as organic acids and sugars were removed, whereas
the relatively hydrophobic fraction was retained. The details on the SPE method using PPL
cartridges and analysis by FTICR-MS can be found in our previous paper (Feng et al., 2016).
Briefly, the mass spectrometry analyses of these samples were performed using a SolariX XR
FT-ICR-MS (Bruker Daltonik GmbH, Bremen, Germany) equipped with a 9.4 T refrigerated
actively shielded superconducting magnet (Bruker Biospin, Wissembourg, France) and a Paracell
analyzer cell. The samples were ionized in positive ion modes using the ESI ion source (Bruker
Daltonik GmbH, Bremen, Germany). A typical mass-resolving power of >400 000 was achieved
at m/z 400 with an absolute mass error of <0.5 ppm. Molecular formulas were assigned to all
ions with signal-to-noise ratios of greater than 10 with a mass tolerance of ±1.5 ppm using
custom software. Molecular formulas with their maximum numbers of atoms were defined as: 30



$^{12}$C, 60 $^1$H, 20 $^{16}$O, 3 $^{14}$N, 1 $^{32}$S, 1 $^{13}$C, 1 $^{18}$O and 1 $^{34}$S. Identified formulas containing
isotopomers (i.e., $^{13}$C, $^{18}$O or $^{34}$S) were not considered. Compounds were detected as either
sodium adducts, $[M + Na]^+$, or protonated species, $[M + H]^-$. We report all detected compounds
as neutral species, unless stated otherwise.

**2.3. Data processing**
The assigned molecular formulas were examined using the van Krevelen diagram (Wu et al.,
2004), double-bond equivalents (DBEs), Kendrick mass defects (KMD) series, and aromatic
indices ($AI_{mod}$). The O/C and H/C ratios were calculated by dividing the number of O and H
atoms, respectively, by the number of C atoms in a formula. DBE analysis was used to determine
the number of rings and double bonds in a molecule. The DBE was calculated using equation 1,
$DBE = 1 + c - h/2 + n/2,$              (1)
where $c$, $h$, and $n$ are the numbers of C, H, and N atoms, respectively, in the formula.

The KMD can be used to search for potential oligomeric units (Hughey et al., 2001). The
Kendrick mass (KM) and KMD for $CH_2$ series were calculated using equations 2 and 3,
$KM = observed\ mass \times 14/14.01565,$          (2)
$KMD = NM - KM,$                   (3)
where 14 is the nominal mass (NM) of $CH_2$, 14.01565 is the exact mass of $CH_2$, and NM is KM
rounded to the nearest integer. A homologous series of compounds differing only by the number
of base units form a horizontal line in a plot of KMD against KM.

$AI_{mod}$ is a measure of the probable aromaticity of a molecule assuming that half the O atoms are
double bonded and half have only σ bonds (Koch and Dittmar, 2006). $AI_{mod}$ was calculated using
equation 4,
$AI_{mod} = (1 + c - 0.5o - 0.5h) / (c - 0.5o - n),$       (4)
where $c$, $o$, and $h$ are the number of C, O, H, and N atoms in the formula. $AI_{mod}$ ranges from 0 for
a purely aliphatic compound to higher values being found for compounds with more double
bonds and that are more aromatic.



## 3. Results and discussions

### 3.1. Chemical characterization of $PM_1$ during F30 and F43 measured by HR-ToF-AMS

The average mass concentration and chemical composition measured by HR-ToF-AMS during F30 and F43 periods were shown in Fig. 1. The mass concentration of $PM_1$ were 9.2 and 10.6 μg $m^{-3}$, respectively, which were at the high range of all filters (1.3 – 10.6 μg $m^{-3}$) because of a continuous long-range transport event at the QOMS (Zhang et al., 2018b). Due to our sample processing error, the mass concentration of filter measured gravimetrically could not be used and thus the fractions of $PM_1$ to $PM_{2.5}$ are not available. However, most of WSOM in $PM_{2.5}$ is in accumulation size mode (less than 1μm) which could be detected by HR-ToF-AMS. The chemical composition of $PM_1$ during F30 and F43 were all dominated by OA (55% and 57%), followed by BC (26% and 22%), sulfate (7% and 8%), nitrate (5% and 6%), and ammonium (5% and 6%). The OA was comprised by biomass burning emitted OA (BBOA), nitrogen-contained OA (NOA), and more-oxidized oxygenated OA (MO-OOA) decomposed by positive matrix factorization (PMF) analysis (Fig. 1). The mass contribution of BBOA was higher during F43 than F30 (32% vs. 22%), whereas the contribution of MO-OOA was higher during F30 than F43 (24% vs. 16%). The mass spectra of OA for these two filter periods were closely similar with a person correlation efficiency ($r$) being 0.9. The elemental ratios of oxygen (O) to carbon (C) of OA were 1.04 and 0.97 for F30 and F43 periods (IA method, Canagaratna et al., 2015), respectively, and accordingly the ratios of hydrogen (H) to C were 1.26 and 1.32. These suggest that the OA during F43 was relatively less oxidized than that during F30 ($t$-test, p<0.05). The six category ions ($C_xH_y^+$, $C_xH_yO_2^+$, $C_xH_yO_1^+$, $C_xH_yN^+$, $C_xH_yO_zN^+$, and $HO^+$) detected by HR-ToF-AMS for these two filter periods were all dominated by $C_xH_yO_2^+$, following by $C_xH_y^+$, $C_xH_yO_1^+$, $C_xH_yN^+$, $C_xH_yO_zN^+$, and $HO^+$. The air mass trajectory analyses using the hybrid single particle Lagrangian integrated trajectory (HYSPLIT) model for F35 and F43 periods show air mass mainly originated from west and southwest of the QOMS across north and northwest India where there were many fire spots during these two periods (Fig. 2). The air mass during F43 was partly (13%) transport with low wind speed and short distance (less than 100 km) indicating some potential fresh OA.



## 3.2. The chemical characteristics of WSOM from ESI-FTICR-MS


A total of 4554 and 5192 molecular formulas was identified by ESI(+)-FTICT-MS over the mass
range of 100-700 Da for F30 and F43, respectively. The identified molecular formulas were
grouped into two subgroups based on their elemental composition, i.e., CHO and CHON, all of
which had equal important contribution (45% – 55%) in number (Fig. 3). Note that individual
species in the ESI-FTICR-MS mass spectra could have many different isomeric structures, then
the percentages reflect only the number of unique molecular formulas and do not reflect the
number of unique molecular formulas in each category. Although there exists the difference on
ionization sensitivity of ESI(+) between different studies, the contribution of CHON in our study
is higher than the results before (10% - 30%) (Laskin et al., 2009; Dzepina et al., 2015). The
distinct contribution of CHON compounds in ESI-FTICR-MS mass spectra is consistent with the
results of HR-ToF-AMS which identified a separate NOA factor in PMF analysis. The mass
spectra of these two samples were highly similar in the distributions of molecular relative
intensity (RI) (Fig. 3). The most abundant peaks were a series of CHO compounds cationized by
$Na^+$ (RI>20%, $C_{19}H_{28}O_3(C_2H_4O)_{n=0-6}Na^+$). These compounds were most likely to contain
carboxylic acid groups that readily form $[M + Na]^+$ ions in the positive mode electrospray
ionization (Smith et al., 2009). The average weighted element ratios of F30 and F45 were 0.43
vs. 0.38 for $O/C_w$, 1.38 vs. 1.33 for $H/C_w$, and 1.72 vs. 1.67 for $OM/OC_w$ (Table 1), suggesting a
relative higher oxidation and saturation degree for F30 than F43. These trends are similar with
those of HR-ToF-AMS results, although the elemental ratios are different between them which is
due to the difference on the detection range of m/z and the ionization efficiency of different mass
spectrometry (ESI vs. EI) (Yu et al., 2016). The elemental ratios of WSOM from ESI-FTICR-
MS in our study are similar with those results observed in aerosol samples in remote site using
ESI-FTICR-MS (e.g., 0.35 – 0.53 for O/C) (Table 2). The CHO compounds had relatively higher
$O/C_w$ ratio than that of CHON compounds in these two samples and CHO compounds were more
saturated with a higher $H/C_w$ ratio than CHON (Table 1). The O/C and H/C in Van Krevelen
diagrams (Wu et al., 2004) for these two filters and the subgroup molecular show similar
distributions and all concentrate in 1.2-1.8 for H/C and 0.3-0.7 for O/C (Fig. 3) suggesting their
similar aerosol sources and atmospheric processes. The similar distributions for these two filters
are also observed in KMD vs. KM plots and located in a narrow and uniform distribution, which


are similar with highly processed aerosol found at the Pico Mountain Observatory (Dzepina et
al., 2015).

Structural information for the assigned molecular formulas is inferred from the $DBE_w$ value
which was higher for F43 than that of F30 (6.92 vs. 6.26) (Table 1). Comparing with other
studies, the DBE values in our filter is relative lower than those in fresh emitted aerosol from
fuel combustion (5 − 9.5) (Song et al., 2018), but close to the results from biomass burning
aerosol and aerosol samples from remote sites (Table 2) (Dzepina et al., 2015). The $DBE_w$ values
for each molecule subgroup were higher for CHON than that of CHO (Table 1), especially for
F43 (7.46 vs. 6.69) suggesting more rings and double bonds in CHON molecular. The $AI_{mod}$,
reflecting the minimum number of carbon-carbon double bonds and rings (Koch and Dittmar,
2006), was correspondingly higher in F43 which contained 49.1% aliphatic (60.4% in F30),
45.9% olefinic (36.8% in F30), and 5.1% aromatic compounds (2.9% in F30). For aromatic
compounds ($AI_{mod}$ >=0.5) in F43, ~60% of them were CHON formulas (39% for F30) (Table 1).
A distinct group of CHON aromatic compounds is shown in lower left corner in Van Krevelen
diagram for F43 but not for F30 (Fig. 3). Higher DBE and $AI_{mod}$ values in CHON compounds
suggest more unsaturated compounds with them which could contain a certain number of
chromophores. The distribution of DBE vs. carbon number of two filters showed a systematic
increase in a concentrated region and a highly similarity with each other. This similarity further
suggests the consistent source and chemical processes for the aerosol of these two filters.

There were 3700 common molecular formulas between these two filters with the number
contribution of CHO by 47% and CHON by 53%. These common molecular formulas accounted
for 81% (F30) and 71% (F43) of two filters, respectively. There were 619 unique molecular
formulas in F30 with 91% being CHO compounds; whereas there were 1142 unique molecular
formulas in F43 with 62% being CHON compounds. For more confidence on molecule
assignment, we focus on the common molecular formulas detected in these two samples in the
section below.



### 3.3. The potential sources and formation processes

### 3.3.1. CHO compounds

CHO compounds have been frequently detected in ambient aerosol samples (Altieri et al., 2009a; Mazzoleni et al., 2010; Lin et al., 2012; Fleming et al., 2018), which could comprise of high molecule weight humic-like substances (HULIS) or oligomers, and from primary emission or secondary formation of different aerosol sources (Mazzoleni et al., 2012; Wozniak et al., 2014; Lin et al., 2016; Cook et al., 2017). In our samples, the weighted molecule weight of CHO compounds was 363.7 with an average C atom of $19.1 \pm 5.3$ per molecule; the most abundant O atoms located in 5-10 with an average value of $7.6 \pm 2.9$ per molecule (Fig. 4a and b). Among the 1744 common CHO formulas, 388 of them (22%) are observed as $[M + Na^+]$ ions with the majority of detected as protonated species. The most abundant sodiated molecules in the ESI/MS ranged in m/z 350-550, whereas the most abundant protonated species ranged in m/z 200-350. The DBE of CHO increased with the carbon number with the $DBE_w$ being 5.96 (Fig. 3); the carbon-normalized $DBE_w$ ($DBE/C_w$) was 0.33 (Table 1). These two values were close to the results from biomass burning aerosol samples in other studies (Table 2) (Lin et al., 2012; Mazzoleni et al., 2012). A threshold DBE/C value of 0.7 usually serves as a criterion to identify species with condensed aromatic ring structures and therefore the CHO compounds in our samples were relative low aromaticity likely due to the relative long oxygenation time during long-range transport. The Carbon oxidation state (OSc) values (Kroll et al., 2011), a useful metric for the degree of oxidation of organic species in the atmosphere, exhibited between -1 and 0 with 25 or less carbon atoms, suggesting that they are semi- and low-volatile organic compounds corresponding to "fresh" (BBOA) and "aged" (LV-OOA) SOA by multistep oxidation reactions (Fig. 4c).

There are several possible sources and chemical formation pathways for high oxygen-containing CHO compounds. Ozonolysis of α-pinene has been found to form highly oxygenated molecules, and one of the products is $C_{17}H_{26}O_8$ (m/z 358) (RI = 9.2%) and $C_{19}H_{28}O_7$ (m/z 368) (RI = 3.2%) which is a possible esterification product of cis-pinic ($C_9H_{14}O_4$, m/z 186, RI = 7.2%) and diaterpenylic acid ($C_8H_{14}O_5$, m/z 190) (Kristensen et al., 2013). The first three compounds were all found in the common CHO molecules with high relative abundance and ionized by $Na^+$.



Ozone concentration in Himalayas during pre-monsoon was highest based on the on-line
measurement at the Nepal Climate Observatory at Pyramid (NCO-P) during 2006-2008 (61 ± 9
ppbv) (Cristofanelli et al., 2010). High biogenic volatile organic compound emissions could
occur due to high density of forest in the southern Himalayas and biogenic secondary organic
aerosol has been found to be important source in Himalayas (Stone et al., 2012). A number of
previously reported other monoterpene oxidation products were also observed in our study, such
as $C_6H_{10}O_5Na^+$ (RI = 1.3%), $C_8H_{10}O_5Na^+$ (RI = 3.9%), $C_8H_{12}O_5H^+$ (RI = 6.7%), $C_9H_{14}O_5H^+$ (RI
= 5.4%), $C_{10}H_{14}O_6Na^+$ (RI = 7.7%), $C_{10}H_{14}O_7Na^+$ (RI = 8.4%), $C_9H_{12}O_6Na^+$ (RI = 3.4%),
$C_7H_{10}O_5Na^+$ (RI = 2.9%), $C_{10}H_{16}O_3H^+$ (RI = 3.8%), $C_7H_{10}O_5Na^+$ (RI = 1.8%), $C_7H_{12}O_5Na^+$ (RI =
2.0%), $C_8H_{12}O_6Na^+$ (RI = 6.7%) (Claeys et al., 2007; Kleindienst et al., 2007; Zhang et al.,
2018a). We also observed formulas that could be lignin pyrolysis products such as vanillic acid
($C_8H_8O_4H^+$, RI = 1.1%), syringaldehyde ($C_9H_{10}O_4H^+$; RI = 2.1%), and syringic acid ($C_9H_{10}O_5H^+$;
RI = 2.7%). In addition, Sun et al. (2010) and Yu et al. (2014; 2016) observed that aqueous-
phase oxidation of lignin produces phenol ($C_6H_6O$), guaiacol ($C_7H_8O_2$) and syringol ($C_8H_{10}O_3$)
yields a substantial fraction of dimers and higher oligomers with key dimer markers identified as
$C_{16}H_{18}O_6$ and $C_{14}H_{14}O_4$. The dimer markers $C_{16}H_{18}O_6H^+$ and $C_{14}H_{14}O_4H^+$ were also present in
our sample with high RI (7.6% and 7.8%) and $C_{16}H_{18}O_6Na^+$ was also observed (RI = 1.5%). The
high relative intensity of these compounds indicate that fog and cloud processing of phenolic
species (biomass burning aerosol) could be an important mechanism for the production of low-
volatility SOA in Himalayas. Some other biomass burning emission compounds were also found,
such as dicarboxylic acid series $C_6H_{10}O_4(CH_2)_nH^+$ and dihydroxycarboxylic acids
$C_9H_{18}O_4(CH_2)_nH^+$; A series of saturated $C_{16}H_{12}O_8(CH_2)_nH^+$ ketones were also observed in the
sample (Fig. 4d). Compounds ($C_9H_{10}O_3H^+$, $C_{10}H_{10}O_3H^+$, $C_{11}H_{10}O_3H^+$, $C_{11}H_{12}O_3H^+$, $C_{12}H_{12}O_2H^+$,
$C_{13}H_{12}O_3H^+$, $C_{13}H_{14}O_3H^+$, $C_{13}H_{14}O_4H^+$, $C_{14}H_{16}O_4H^+$, $C_{14}H_{16}O_3H^+$) observed in biomass burning
emission (cow dung and brush wood) sampling during residential cooking in Nepal (Fleming et
al., 2018) were also found in our samples.

**3.3.2 CHON compounds**
The frequency distribution for $n_o$ and $n_c$ in CHON formulas were shown in Fig. 5a which show
peaks between 6-10 and 15-20, respectively. The DBE of CHON formulas ranged into 4 – 10





with $DBE_w$ being 6.79 (Fig. 5b). In the $CHON^+$ class, compounds contained one or two nitrogen
(1N or 2N) atoms with 1N compounds accounting for 70.5% and 2N for 29.5%, respectively.
Most (98%) of 1N compounds contained more than 3 oxygen atoms and could up to 13 oxygen
atoms, whereas about 62.5% of 2N compounds contained more than 6 oxygen atoms (Fig. 6a).
The average O atom containing in each molecular formula was therefore higher for 1N
compounds than 2N compounds ($8.1 \pm 2.9$ vs. $6.3 \pm 2.3$). The high O atom containing in CHON
formula suggest that nitrogen was in the form of organic nitrate or nitro groups with excess
oxygen forming additional oxygenated groups. The ratios of $O/C_w$ and $OSc_w$ for 1N compounds
were accordingly higher than that of 2N compounds (0.42 vs. 0.37 for $O/C_w$; $-0.48$ vs. $-0.54$ for
$OSc_w$), suggesting higher oxidation state for 1N compounds (Fig. 5c). In contrast, the $DBE_w$ and
$AI_{mod}$ values for 2N compounds were higher than that of 1N compounds (Table 1). With higher
$H/C_w$ for 2N compounds (Table 1), it suggests that 2N compounds could contain amount of
aromatic N-heterocyclic compounds. For 1N compounds, longer and higher relative intensity
$CH_2$ homologous series compounds were found based on the Kendrick mass defect plot (Fig.
6b); 1073 of the 1378 detected 1N compounds can be grouped into 145 homologous. The
abundant long $CH_2$ homologous series in 1N compounds contained 7-10 O atoms, while 5-8 O
atoms for 2N compounds (Fig. 6).

There are many potential sources for CHON compounds, such as amino acids, reduced N
compounds, nitro compounds, and organic nitrate (Altieri et al., 2009b; Laskin et al., 2009;
O'Brien et al., 2013). Biomass burning has been found to be an important source for nitrogen-
containing compounds in atmosphere (Hoffmann et al., 2007). Laskin et al. (2009) identified
amount of N-heterocyclic alkaloid compounds from kinds of fresh biomass burning aerosol.
Fleming et al. (2018) conducted study in Nepal by collecting fresh emitted aerosol from dung
and brushwood burning household cookstoves and identified amount of nitrogen-containing
aerosol. Oxygenated organic nitrogen compounds in ambient aerosol (Dzepina et al., 2015), rain
water (Altieri et al., 2009b), and fog water (Mazzoleni et al., 2010) from biomass burning
emission influenced regions were also observed. Although the dominated CHN compounds in
fresh aerosol in Laskin et al. (2009) and Fleming et al. (2018) were not found in our study which
was likely due to the highly oxygenated OA in our samples, biomass burning emissions could be



an important source for CHON compounds in our study. Recent studies have proven that burning
of dung-fuel in Nepal can emit amount of nitrogen species such as $NH_3$, $NO_x$, HCN, benzene,
and organics, and the emission factors for these species are higher than that of wood (Stockwell
et al., 2016; Jayarathne et al., 2018). In addition, it is likely that smoldering burning of bio-fuels
in high elevation area is also responsible for the presence of a large number of nitrogen-
containing compounds in BBOA. Nitroaromatic compounds such as Methyl-Nitrocatechols
($C_7H_7NO_4$) are introduced to be tracer for biomass burning secondary organic aerosols (Iinuma
et al., 2010). Although $C_7H_7NO_4$ formula is not found in our measurement, $C_{14}H_{14}N_2O_8$ were
found in our measurement, of which is probably its dimer formula. In addition, the homologous
series compounds which $C_7H_7NO_4$ serve as the core molecule was also found in our samples.
Some medium relative abundance molecular formulas identified in a recent paper in biomass
burning aerosol were also found in our measurement such as $C_{15}H_{19}N_1O_8H^+$ (RI = 6.7%),
$C_{16}H_{21}N_1O_8H^+$ (RI = 6.6%), $C_{17}H_{23}N_1O_8H^+$ (RI = 5.9%), $C_{14}H_{13}N_1O_3H^+$ (RI = 2.9%),
$C_{15}H_{15}N_1O_3H^+$ (RI = 2.6%), $C_{13}H_{17}N_1O_3H^+$ (RI = 4.2%) (Song et al., 2018).

Besides primary emission and/or secondary formation from biomass burning emission, nitrogen-
containing OA could also be formed through other chemical processes. For example, biogenic
volatile organic compounds (BVOC) can react with $NO_3$ radical or $RO_2$+NO into organic nitrate
(Ng et al., 2017). Although organic nitrate is not favored to be ionized in positive ESI-MS (Wan
and Yu, 2006), organic nitrate formed from BVOC could be highly functionalized (Lee et al.,
2016) and ionized in positive MS through other alkaline functional groups. Recent studies have
shown that BVOC, including isoprene ($C_5H_8$) and monoterpenes ($C_{10}H_{16}$), dominate the organic
nitrate formation in the southeastern United States under the condition of the mixed
anthropogenic $NO_x$ and BVOC (Xu et al., 2014; Lee et al., 2016; Zhang et al., 2018a). Several
molecular formulas formed from monoterpene and $NO_3$ radical were found in our study
($C_9H_{13}NO_6$, $C_9H_{15}NO_6$, $C_{10}H_{17}NO_4$, $C_{10}H_{15}NO_5$, $C_{10}H_{17}NO_5$, $C_{10}H_{19}NO_5$, $C_{10}H_{15}NO_6$,
$C_{10}H_{17}NO_6$, $C_{10}H_{19}NO_6$,) (Lee et al., 2016; Zhang et al., 2018a). Additionally, some alkaloids,
such as imidazole, imidazole-2-carboxaldehyde and 1N-glyoxal-substittuted imidazole, are also
reported to be major products of BVOC reaction with ammonium ions or primary amines on



SOA (De Haan et al., 2009; Galloway et al., 2009; Updyke et al., 2012). Imidazole signal was
found in our HR-ToF-AMS measurement.

**4. Implications**
This study analyzed the WSOM using ESI(+)-FTICR-MS in fine particular aerosol from
Himalayas and found that the molecular compositions of WSOM were mainly comprised by
CHO and CHON compounds with equal important contribution. The two compounds could be
originated from biomass burning emission and BVOC oxidation of which many markers were
found in these molecular compounds. All our compounds had relatively high DBE values
suggesting potential high light absorption feature. Due to their relative high mass concentration
and high contribution of nitrogen-containing compounds (7.6% of $PM_1$) based on HR-ToF-AMS
results, it is believe that the aerosol transported to Himalayas have important application in
atmospheric radiative forcing and biogeochemical effects.

Ramanathan and Carmichael (2008) found distinct warming effect of light absorbing aerosol
over Himalayas through estimating aerosol radiative forcing by BC. However, light-absorbing
OA (brown carbon) could also be another important light absorbing aerosol due to their large
fraction of atmospheric aerosol loading (Laskin et al., 2015). Zhang et al. (2017) estimated the
light absorption contribution of brown carbon from inland of the TP which was up to ~13% of
that of BC. The high DBE and nitrogen-containing OA in our study suggested aerosol in
Himalayas could contain amount of light-absorbing organic matter because light absorption
properties of organic molecules are closely related with the number of double bonds and rings in
the molecule and nitrogen atoms. Many studies had found that the dominated chemical
molecules in the brown carbon were related with nitrogen-containing aerosol (e.g., Lin et al.,
2016). This kind of aerosol combined with BC could have higher radiative forcing than before in
this area.

In addition, nitrogen is an important nutrient for plant and microbial in terrestrial and aquatic
systems especially for arid and semi-arid areas (Andreae and Crutzen, 1997). Chen et al. (2013)
evaluated the potential biogeochemical cycle in the TP under the current rapidly climate change





and suggested that nitrogen availability plays a critical role in controlling ecological production
because nitrogen is a limited nutrient in many ecosystems. To our knowledge, there was no study
focusing on the organic nitrogen deposition in this remote region, but only on inorganic nitrogen
species (Liu et al., 2015).  Organic nitrogen is an important source of nitrogen (Neff et al., 2002),
and should be taken account in the future study.

**Acknowledgements**
This research was supported by grants from the National Natural Science Foundation of China
(41771079, 41421061), the Key Laboratory of Cryospheric Sciences Scientific Research
Foundation (SKLCS-ZZ-2018), and the Chinese Academy of Sciences Hundred Talents
Program. The authors thank the QOMS for logistic support.

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





Tables 1. Chemical characterization of the molecular assignments detected in WSOM for F30,
F43, and common ions. Relative intensity weighted (w) each data subset (O/C, H/C, OM/OC,
DBE, and DBE/C) are given.

| | | All | CHO | CHON |
|---|---|---|---|---|
| F30 | $O / C_w$ | 0.43 | 0.43 | 0.42 |
| | $H / C_w$ | 1.38 | 1.40 | 1.35 |
| | $OM / OC_w$ | 1.72 | 1.69 | 1.77 |
| | $DBE_w$ | 6.26 | 6.00 | 6.69 |
| | $DBE / C_w$ | 0.34 | 0.32 | 0.36 |
| | Number of Aliphatic (AI = 0) | 2602 | 1514 | 1088 |
| | Olefinic (0.5 > AI > 0) | 1584 | 708 | 876 |
| | Aromatic (AI >= 0.5) | 123 | 75 | 48 |
| F43 | $O / C_w$ | 0.38 | 0.38 | 0.39 |
| | $H / C_w$ | 1.33 | 1.35 | 1.31 |
| | $OM / OC_w$ | 1.67 | 1.62 | 1.72 |
| | $DBE_w$ | 6.92 | 6.36 | 7.46 |
| | $DBE / C_w$ | 0.36 | 0.34 | 0.38 |
| | Number of Aliphatic (AI = 0) | 2413 | 1200 | 1213 |
| | Olefinic (0.5 > AI > 0) | 2256 | 870 | 1306 |
| | Aromatic (AI >= 0.5) | 249 | 103 | 146 |
| Common ions | $O / C_w$ | 0.41 | 0.43 | 0.40 |
| | $H / C_w$ | 1.36 | 1.34 | 1.37 |
| | $OM / OC_w$ | 1.70 | 1.65 | 1.76 |
| | $DBE_w$ | 6.33 | 5.96 | 6.79 |
| | $DBE / C_w$ | 0.34 | 0.33 | 0.37 |
| | Number of Aliphatic (AI = 0) | 2067 | 1019 | 1048 |
| | Olefinic (0.5 > AI > 0) | 1518 | 653 | 865 |
| | Aromatic (AI >= 0.5) | 112 | 71 | 41 |








Table 2. Chemical characterization of the molecular assignments detected in aerosol samples
from selected studies (adapted and modified from Table 3 in Dzepina et al. (2015)). Note that all
values are presented as arithmetic mean.

| Sample type | Measurement site | Instrument | O / C | H / C | OM / OC | DBE | DBE / C | Reference |
|---|---|---|---|---|---|---|---|---|
| Aerosol | Free troposphere | ESI(+)-FTICR-MS | 0.38–0.42 | 1.27–1.31 | 1.66–1.71 | 7.73–8.62 | 0.36–0.37 | This study |
| Aerosol | Free troposphere | ESI(−)-FTICR-MS | 0.42–0.46 | 1.17–1.28 | 1.67–1.73 | 9.4–10.7 | 0.42–0.47 | Dzepina et al. (2015) |
| Aerosol | Free troposphere | ESI(−)-FTICR-MS | 0.53 ± 0.2 | 1.48 ± 0.3 | 1.91 ± 0.3 | 6.18 ± 3.0 | / | Mazzoleni et al. (2012) |
| Aerosol | Rural | ESI(−)-FTICR-MS | 0.28–0.32 | 1.37–1.46 | 1.54–1.60 | 6.30–7.45 | 0.33–0.38 | Wozniak et al. (2008) |
| Aerosol | Suburban | ESI(−)-FTICR-MS | 0.46 | 1.34 | 1.85 | 5.3 | 0.45 | Lin et al. (2012) |
| Aerosol | Urban | ESI(+)-FTICR-MS | 0.31 | 1.34 | / | 8.68 | 0.41 | Choi et al. (2017) |
| Aerosol | Marin boundary layer | ESI(−)-FTICR-MS | 0.35 | 1.59 | 1.67 | 4.37 | 0.28 | Schmitt-Kopplin et al. (2012) |
| Aerosol | Marine boundary layer | ESI(−)-FTICR-MS | 0.36 –0.42 | 1.49–1.56 | 1.70–1.74 | 5.88–6.76 | 0.28–0.32 | Wozniak et al. (2014) |
| Cloud water | Free troposphere | ESI(−)-FTICR-MS | 0.61–0.62 | 1.46 | 2.06–2.08 | 6.29–6.30 | 0.38 | Zhao et al. (2013) |
| Cloud water | Rural | ESI(−)-FTICR-MS | 0.51 | 1.47 | / | 6.03 | / | Cook et al. (2017) |
| Fog water | Rural | ESI(−)-FTICR-MS | 0.43 | 1.39 | 1.77 | 5.6 | 0.40 | Mazzoleni et al. (2010) |








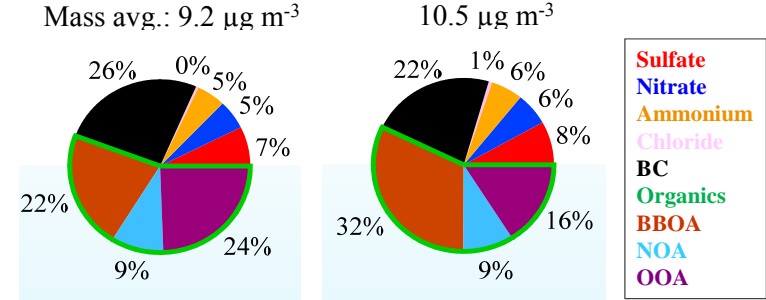


Fig. 1. The average mass concentration and chemical composition of $PM_1$ during F30 (left) and
F43 (right) periods, respectively, measured by HR-ToF-AMS and PAX.






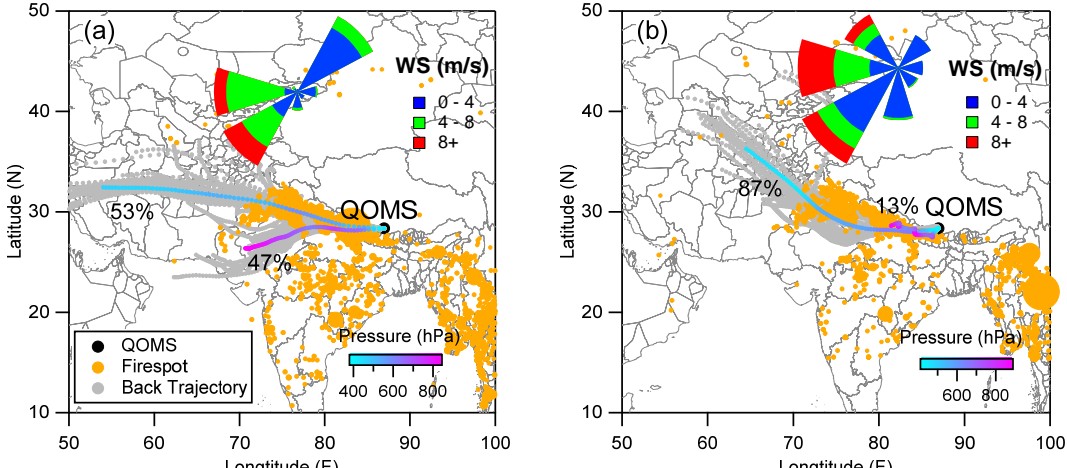


Fig. 2. The air mass back trajectory analysis using HYSPLIT model (Draxler and Hess, 1998) during (a) F30 and (b) F43. The air mass trajectories were recovered back to 72 h at 1 h interval from the sampling site (QOMS) above the ground level of 1000 m using 1° resolution Global Data Assimilation System (GDAS) dataset (https://ready.arl.noaa.gov/gdas1.php). The cluster analysis for these trajectories was completed based on the directions of the trajectories (angle distance) and colored according to air pressure. The fire spot observed from MODIS (https://firms.modaps.eosdis.nasa.gov) and the average wind rose plot for during each filter sampling period were also shown. The fire spot is sized by fire radiative power (FRP).






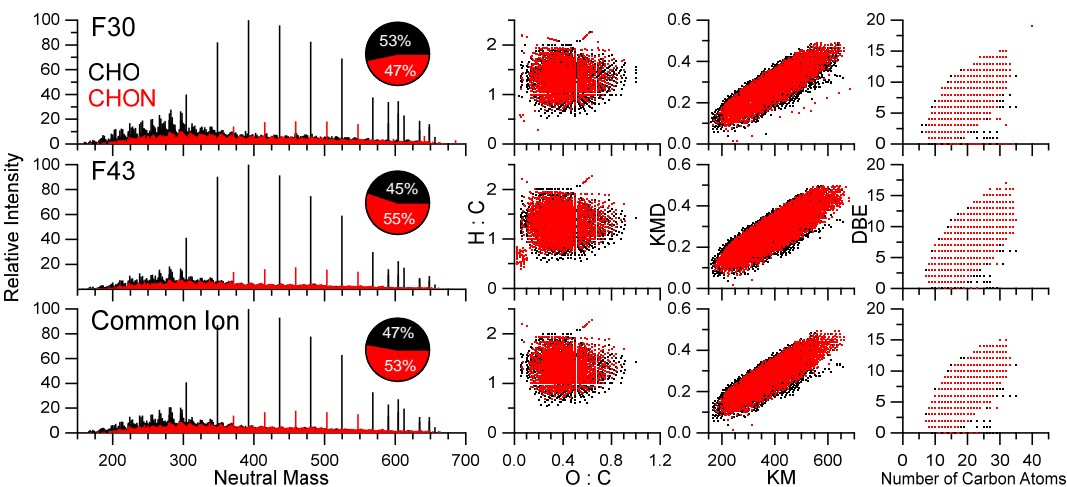


Fig. 3. The combo plot for F30, F43, and Common Ion including high-resolution mass spectrum,
Van Krevelen diagram, KMD vs. KM, and DBE vs. number of carbon atoms.




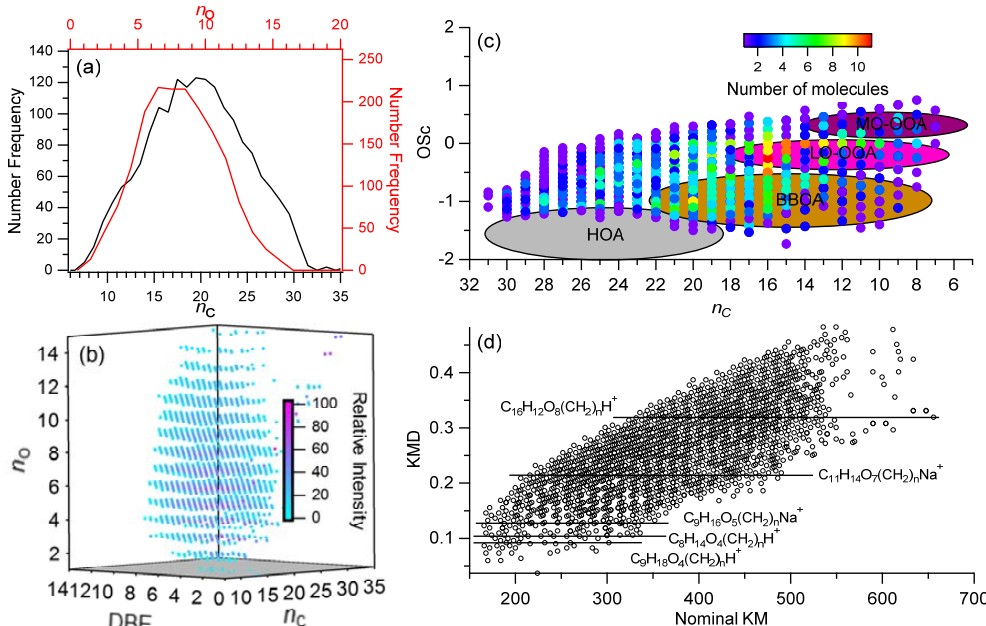


Fig. 4. The molecular information for common CHO compounds of two filters. (a) The number
frequency distribution of carbon ($n_c$) and oxygen ($n_o$); (b) The 3-D plot for $n_o$, $n_c$, and DBE
colored by their relative intensity; (c) Scatter plot of carbon based oxidation state (OSc) vs. nc
colored by the distribution of number of molecules; (d) The Van Kerevelen diagram by H/C vs.
N/C colored by no. The size of dot marker in (c) and (d) represent the 1N and 2N compounds.



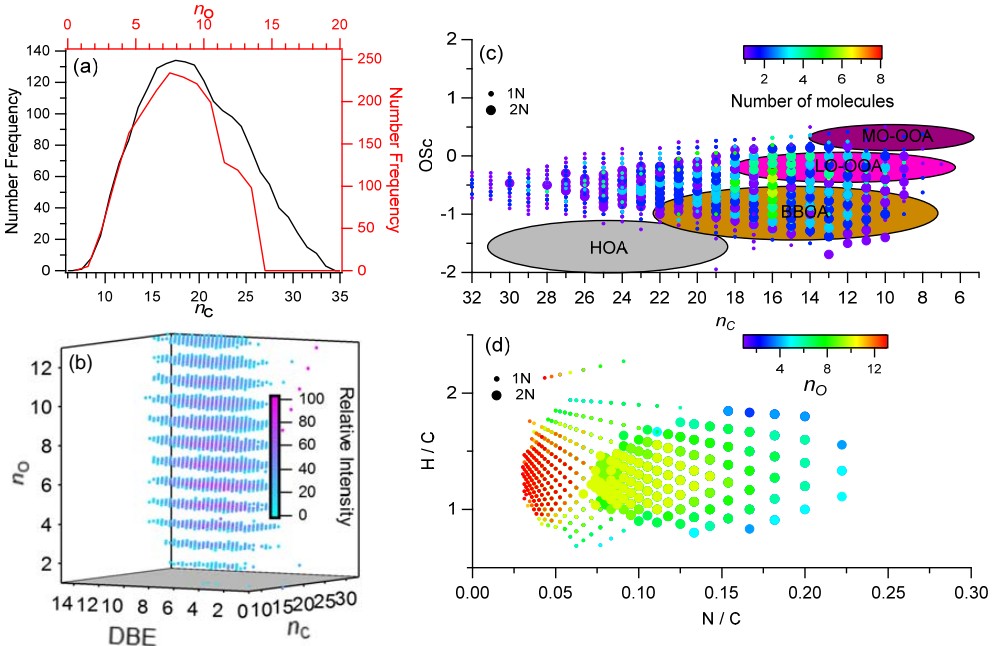


Fig. 5. The molecular information for common CHON compounds of two filters. (a) The number
frequency distribution of carbon ($n_c$) and oxygen ($n_o$). (b) The 3-D plot for $n_o$, $n_c$, and DBE
colored by their relative intensity. (c)  Scatter plot of carbon based oxidation state (OSc) vs. nc
colored by the distribution of number of molecules. (d) The Van Kerevelen diagram by H/C vs.
N/C colored by no. The size of dot marker in (c) and (d) represent the 1N and 2N compounds.





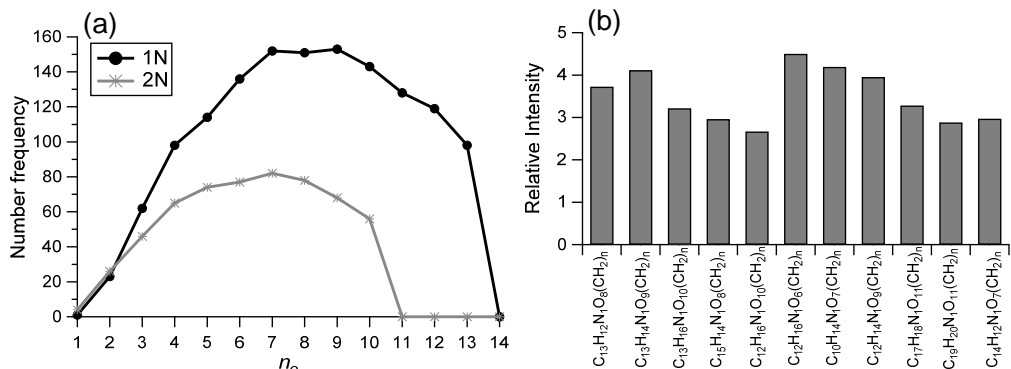


Fig. 6. (a) The number frequency distribution of $n_o$ for 1N and 2N compounds and (b) the longest
ten CH2 homologous series compounds in 1N compounds.