# Peer review of "Molecular characterization of organic aerosol in Himalayas: insight from"

_Atmospheric Chemistry and Physics, 2018_

## Referee Comment (RC1) · Anonymous Referee #1 · 4 Sep 2018

This study analyzed the molecular chemical composition of water soluble organic matter (WSOM) from two fine particulate filter samples collected at a high altitude station (Qomolangma Station, QOMS, 4276 m a.s.l.) in the northern Himalayas using positive mode electrospray ionization Fourier transform ion cyclotron resonance mass spectrometry (ESI(+)-FTICR-MS). The molecular compositions of WSOM mainly comprised CHO and CHON compounds with equal important contribution. Detailed molecular information in the common formula of these two filters was explored. The authors found that water-soluble organic compounds were mainly from biomass burning and biogenic emissions. All compounds had relatively high DBE values suggesting potential high light absorption feature and have important application in atmospheric radiative forcing

and biogeochemical effects in the remote region. As the analysis of molecular chemical compositions of WSOM using ultra-high resolution mass spectrometry in such a high altitude regions is rare and important, the data set provided by this work is thus very valuable. The authors also performed a comprehensive analysis on this dataset, and the findings, conclusions are well supported by such analyses. Overall, the paper is within the scope of ACP and generally well written and documented. I recommend publication of this paper in ACP after some revisions.

Specific comments: (1) Line 19, the weighted double bond equivalent (DBEw) was used here and in Table 1, however, the calculation method for DBEw was not given in Sect. 2.3 besides that for DBE, please added. (2) Line 69-76, the advantages of FTICR-MS method compared with the previous measurements in HTP as well as the wide usage of FTICR-MS worldwide need to be more emphasized in the introduction, whereas the current version were relatively simple. (3) Line 82-93, the logic in these sentences about the description of sampling site and instruments are confused, namely the sentence of "and the instruments used in this study...BC mass concentration" need to be moved before "A low-volume (16.7 L min-1)...". Overall, the description of sampling site and weather first following by the instrument. Besides, the instruments used in this study included a HR-ToF-AMS, PAX, and PQ-200, rather than just HR-ToF-AMS and PAX but description PQ-200 alone in the following part. (4) Line 163, "However, most of WSOM in PM2.5 is in accumulation size mode (less than 1 $\mu$m) which could be detected by HR-ToF-AMS.", please provide reference. (5) Line 190-191, please rephrase this sentence and make it easy to understand. (6) Line 197, the common ions are selected from the two samples in Fig. 3 and Table, however, how to calculate the RI for these common ions? From F43? please verify. (7) Line 202, "suggesting a relative higher oxidation and saturation degree" is different from that in Line 20 in abstract of " suggesting their medium oxidation and saturation degrees." (8) Line 205, EI is first mentioned here in the manuscript, please add the full description. (9) Line 207-208, the statement of "The CHO compounds had relatively higher O/Cw ratio than that of CHON compounds in these two samples" is inappropriate for F43 in Table 1,

please check. (10) Line 218-221, the author compared the DBR values (7.73-8.62 from Table 2) in this study with those in previous studies, however, the comparisons were not clear as the author declared that the DBE values is relatively lower than 5 – 9.5 (Song et al., 2018), but close to 9.4 – 10.7 (Dzepina et al., 2015), please rephrase. (11) Line 225-226, "The AImod, ..., was correspondingly higher in F43 which contained 49.1% aliphatic (60.4% in F30), 45.9% olefinic (36.8% in F30), and 5.1% aromatic compounds (2.9% in F30)." What was correspondingly higher in F43? The total number of the following three compounds? Consideration the higher number in F30 for aliphatic, the current expression is ambiguous. (12) Line 237-279, a total of 4554 and 5192 molecular formulas was identified for F30 and F43 and existed 3700 common molecular formulas, however, the unique molecular formulas were just 619 and 1142 for the two filter rather than the rest 4554-3700 and 5192-3700, please modify or add specific values to Table 1. (13) Line 258, "A threshold DBE/C value of 0.7 usually serves as a criterion to identify species with condensed aromatic ring structures", please added references. (14) Line 269-270, the RI values mentioned here are in this study rather than in the reference, please declare. (15) Line 313, there is no information about the 1N and 2N compounds in the Table 1. (16) Line 372, the 7.6% contribution of nitrogen-containing compounds to PM1 is from Zhang et al. (2018) for the entire long period rather than the two filter period (9% in Fig. 1), please added the reference to give a clear description.

Minor comments: (1) Line 18, change "are" to "were" (2) Line 22, change "significant" to "significantly" (3) Line 25, change "diagnose" to "diagnostic" (4) Line 26, change "highly" to "high" (5) Line 29, change "biomass mass burning" to "biomass burning" (6) Line 31, remove "are" (7) Line 53, change "could" to "can" (8) Line 57, remove "of", change "elevation" to "elevated" (9) Line 84, change "includes" to "included" (10) Line 92, change "with an average temperature..." to "and an average temperature..." (11) Line 101, 219, change "relative" to "relatively" (12) Line 218, change "than than of F30" to "than that for F30" (13) Line 260, change "were" to "have" (14) Line 366, change "particular" to "particulate"

---

## Referee Comment (RC2) · Anonymous Referee #2 · 5 Sep 2018

The authors of this paper reported sampled water-soluble organic matter (WSOM) at a high elevation site of Himalayas. They identified CHO and CHON compounds to be primary molecular compositions of WSOM. The paper provided important information on organic aerosols which could contribute to climate change and atmospheric oxidation over the Tibetan Plateau. The article should be published in ACPD. At this stage, I have only several minor comments as outlined below.

The authors attributed the sources of WSOM to biogenic volatile organic compounds and biomass mass burning. Given the lower temperature ($\sim$5.7 °C), BVOCs might not be readily formed and unlikely transported from distance sources as well. Or there is a

[Figure]

high level of solar radiation/photosynthetically active radiation (PAR) in the Himalayas which favors the BVOC formation? Authors perhaps need to make a comment on this point.

I don't think that the measurement site of this study was located in the free troposphere (Table 2ïïjŇand perhaps the sampling sites in other studies listed in Table 2). The free troposphere means the troposphere above the boundary-layer. Although the elevation of the sampling site is 4275 m, it is on the ground surface and hence within the boundary layer.

Please check the language carefully. There are quite a lot of grammar and spelling errors. For example, line 51, 'is' should be 'are'; line 53, 'could from'; line 57, 'in the southern of'; line 158 and 164, 'were' should be 'was', line 171, 'person correlation efficiency'; line 181, 'transport' should be 'transported'; line 698 (Fig. 2 captions), 'ground level of 1000m', you mean 1000 m above the ground level? These are part of language errors.

---

## Referee Comment (RC3) · Anonymous Referee #3 · 20 Sep 2018

The manuscript presented by Y. An et al. presents the detailed molecular chemistry of two samples collected from the Qomolangma Station in the Himalayas. The detailed molecular chemistry was derived from ultra-high resolution FT-ICR mass spectrometry measurements following electrospray ionization to generate positive ions. The authors discuss the molecular composition of the two samples and compare them carefully to previously published studies using similar approaches. Specifically, they found an increased degree of unsaturation of the prominent species in their study. As the authors suggest, these molecules may contribute to aerosol absorption.

The manuscript is well prepared and the methodology is technically sound. However,

[Figure]

I recommend the authors consider revisions to the manuscript to address method limitations pertaining to the ionization (potential artifacts, differences between + and -, anticipated functional groups, etc.) and discuss more specifically the significance of the results (both with respect to the implications and the limitations).

Specific major comments without any predetermined order:

1. It appears that many assumptions about the ionization method were made in the data interpretation. Those assumptions are not explicitly stated and may be incorrect. (i) For example, NH4+ is a common cation that readily adducts to molecular to assist in forming positive ions (similar to Na+). Please explicitly state your assumptions regarding this possible artifact. (ii) CHON compounds observed in ESI are expected to vary with the ionization mode. For example, reduced N (e.g., amino functional groups) are not expected to be observed in the negative ion mode. Likewise, oxidized N (e.g., nitrate functional groups) are not expected to be observed in the positive ion mode. Please explicitly state your assumptions regarding the ionization method and possible differences between ESI positive ions and ESI negative ions.

2. Due to differences in the ESI ionization process (positive vs negative), the direct comparison of the data can be difficult. Please be sure to check the ionization mode of referenced datasets and discuss the method limitations associated with the datasets and the resulting limitations on the conclusions.

3. How were the molecular formulas and their homologous series formed from biogenic VOCs and biomass burning identified?

4. The phrase "important implications" (line 32) is an empty phrase. Please be more specific with the inferred implications and impacts associated with the studied molecular classes.

5. What do your sample names indicate or represent? Consider changing the samples names to be more descriptive.

6. What is meant by "pristine region"?

7. The literature review describing the significance of light absorbing aerosol is severely out of date.

8. The phrase "Many studies" requires more than just one example reference.

9. The goal of the paper is what exactly?

10. What is the relevance of discussing the tourist season?

11. Ultrasonic baths can introduce reactive oxygen species. What care was taken to avoid extraction artifacts?

12. Please discuss the SPE recovery.

13. Please discuss the steps that were taken to avoid ESI artifacts?

14. The parameters associated with your "custom software" need to be more thoroughly described. How did you eliminate ambiguous formula assignments?

15. What is meant by the "processing error" mentioned in line 161? How did you ensure that the error did not affect the molecular composition?

16. What is the procedure for identifying the NOA compounds with HR-ToF-AMS?

17. How was the influence of potential fresh OA inferred?

18. In the discussion of common ions, the authors assume that the two samples have a similar aerosol source? What if instead, the common compounds are simply not marker compounds. Complex mixtures are expected to have many ions in common.

19. What is the balance of source contribution vs. aerosol aging in these samples.

20. I'm surprised that the long-range transported aerosol reported in Dzepina et al. is similar to the samples reported here. How is that observation justified with respect to the transport pathways?

21. The "distinct group of CHON aromatic compounds" in the lower left of the VK diagram may be incorrect assignments. What are the limits for the DBE range?

22. What is the significance of the difference in the max abundance between H+ and Na+ type ions?

23. The detailed description given over lines 279-298 is quite tedious. Perhaps some of these formulas can be better presented in a figure or table?

24. What is the significance of 1N vs 2N?

25. How do you observe acidic N in the positive ion mode?

26. The statement in lines 330-333 is not convincing. Please rephrase and add more evidence or description.

27. Lines 351- 353: How does the sample matrix effect the observation of ions in ESI?

28. Where are the major products of BVOC as mentioned in lines 359-363?

29. The discussion of the research implications can be enhanced with a deeper discussion of the molecular composition and method limitations. What other observations of absorbing species have been made in the Himalayas?

30. The implications regarding nutrients and biogeochemical cycling are beyond the scope of the current research and seem to be a bit too ambitious. Please revise.

31. Related to the previous comment, did you study deposition?

32. Again, what type of N did you study with your analytical method. Please be clear with the limitations and assumptions that are necessary.

33. How are the measurement sites defined? The listed free troposphere sites do not consistently sample free tropospheric air. In each case, seasonal factors may play a strong role in the height of the boundary layer.

34. Figure 3 appears to contain several high intensity regularly spaced peaks that are

not associated with the sample. Please remove or flag these peaks as contaminants.

35. Please add the specific details regarding your treatment of blank samples to the methods section.

36. How are the FT-ICR MS ions related to the fragment ions from HR-ToF-AMS (as shown in figure 4)?

Additional minor comments:

1. Line 19: DBE = double bond equivalents; DBE is plural not singular

2. Line 31: "high nitrogen containing of aerosol" is unclear. Please rephrase.

3. Line 32: "important implications" (use plural)

4. Line 32: "and the biogeochemical cycle" (insert article)

5. Line 42: "Accompany" is awkward. Please rephrase.

6. Line 43: "to the Himalayas" (insert article)

7. Line 46: "essential" is awkward. Please rephrase

8. Line 172: What is the IA method?

9. Line 181: "transported" (verb tense)

10. Line 185: Fix typo.

11. Lines 247 & 249: "molecular weight" not "molecule weight"

12. Lines 256 & 258: Typo? Did you mean to use Cw or C in these two sentences?

13. Line 261: "carbon oxidation state" (lower case "C")

14. Line 307: "average O atoms contained in" (plural and verb tense)

15. Line 368: "two compounds" or "two compound groups"

16. Line 373: "believed" (verb tense)

17. Table 1 (and Figure 3): Are the values shown for F30 and F43 for all ions or only unique ions?

18. Figure 1: What is the purpose of the blue shading behind the pie chart?

19. Figure 1 (and elsewhere): Please remember to define all of the acronyms used in the figure within the figure caption.

20. Figure 2: Please include the vertical profile for the back trajectories.

---

## Author Comment (AC1) · 11 Nov 2018

**Response to reviewers' comments**

We appreciate the comments and suggestions from three reviewers. We have carefully considered these comments and revise the manuscript accordingly. Our point-to-point responses are presented below. Note that the comments are in black, our response in blue, and revise in the manuscript in red.

ACPD manuscript: **10.5194/acp-2018-693**
Authors: **Yanqing An, Jianzhong Xu, Lin Feng, Xinghua Zhang, Yanmei Liu, Shichang Kang, Bin Jiang, Yuhong Liao**

**Reviewer #1**

This study analyzed the molecular chemical composition of water soluble organic matter (WSOM) from two fine particulate filter samples collected at a high altitude station(Qomolangma Station, QOMS, 4276 m a.s.l.) in the northern Himalayas using positive mode electrospray ionization Fourier transform ion cyclotron resonance mass spectrometry (ESI(+)-FTICR-MS). The molecular compositions of WSOM mainly comprised CHO and CHON compounds with equal important contribution. Detailed molecular information in the common formula of these two filters was explored. The authors found that water-soluble organic compounds were mainly from biomass burning and biogenic emissions. All compounds had relatively high DBE values suggesting potential high light absorption feature and have important application in atmospheric radiative forcing and biogeochemical effects in the remote region. As the analysis of molecular chemical compositions of WSOM using ultra-high resolution mass spectrometry in such a high altitude regions is rare and important, the data set provided by this work is thus very valuable. The authors also performed a comprehensive analysis on this dataset, and the findings, conclusions are well supported by such analyses. Overall, the paper is within the scope of ACP and generally well written and documented. I recommend publication of this paper in ACP after some revisions.

Specific comments:
1. Line 19, the weighted double bond equivalent (DBEw) was used here and in Table 1, however, the calculation method for DBEw was not given in Sect. 2.3 besides that for DBE, please added.

The equation for calculating DBEw and other weighted indexes is added in the method section as follows.

"The wighted DBE (DBEw), O/C (O/Cw), and H/C (H/Cw) were calculated using equation 2,
$$X_w = \sum(w_i * X_i) / \sum w_i \qquad (2)$$
where $X_i$ and $w_i$ are the parameters above and the relative intensity (RI) for each individual formula, i."

2. Line 69-76, the advantages of FTICR-MS method compared with the previous measurements in HTP as well as the wide usage of FTICR-MS worldwide need to be more emphasized in the introduction, whereas the current version were relatively simple.

We have enriched this part in the updated manuscript as follows.

"Fourier transform-ion cyclotron resonance mass spectrometry (FTICR-MS) coupled with soft ionization source, such as electrospray ionization (ESI), can be used to identify the individual molecular formula of extremely complex mixture because of its ultra-high resolution and mass accuracy (Mazzoleni et al., 2010). Similar methods have been used for identification of components in aqueous secondary OA (SOA) and in ambient samples, and allow the identification and separation of thousands of compounds in a sample (e.g., Mazzoleni et al., 2010; Altieri et al., 2012; Mead et al., 2013). Kinds of methods such as double bond equivalents (DBE), elemental ratios, Kendrick mass defects (KMD) can be applied to deduce the chemical characterization of obtained molecular."

3. Line 82-93, the logic in these sentences about the description of sampling site and instruments are confused, namely the sentence of "and the instruments used in this study...BC mass concentration" need to be moved before "A low-volume (16.7 L min-1)...". Overall, the description of sampling site and weather first following by the instrument. Besides, the instruments used in this study included a HR-ToF-AMS, PAX, and PQ-200, rather than just HR-ToF-AMS and PAX but description PQ-200 alone in the following part.

The logic of this paragraph has been updated as suggested by the reviewer.

4. Line 163, "However, most of WSOM in PM2.5 is in accumulation size mode (less than 1μm) which could be detected by HR-ToF-AMS.", please provide reference.

One related literature is added in the updated manuscript.

"Zhang, Q., Canagaratna, M.R., Jayne, J.T., Worsnop, D.R., Jimenez, J.L., 2005. Time- and size-resolved chemical composition of submicron particles in Pittsburgh: Implications for aerosol sources and processes. J. Geophys. Res., 110, D07S09, 10.1029/2004JD004649."

5. Line 190-191, please rephrase this sentence and make it easy to understand.

Revised as follows.

"Note that individual species in the ESI-FTICR MS mass spectra could have many different isomeric structures, then the percentages reflect only the number of unique molecular formulas in each category."

6. Line 197, the common ions are selected from the two samples in Fig. 3 and Table, however, how to calculate the RI for these common ions? From F43? please verify.

The RI for the common ions was calculated from the average of RI in two spectra and then normalized to the highest peak. We added this information in the 3.2 section as follows.

"Note that the mass spectrum of common ions was calculated from the average RI of the individual common ion from two mass spectra and normalized to the highest peak."

7. Line 202, "suggesting a relative higher oxidation and saturation degree" is different from that in Line 20 in abstract of " suggesting their medium oxidation and saturation degrees."

We have revised these two sentences to be consistent as follows.

"…suggesting higher oxidation and saturation degrees for P1."

8. Line 205, EI is first mentioned here in the manuscript, please add the full description.

Revised as the reviewer suggested.

9. Line 207-208, the statement of "The CHO compounds had relatively higher O/Cw ratio than that of CHON compounds in these two samples" is inappropriate for F43 in Table 1, please check.

The sentence has been removed.

10. Line 218-221, the author compared the DBR values (7.73-8.62 from Table 2) in this study with those in previous studies, however, the comparisons were not clear as the author declared that the DBE values is relatively lower than 5 –9.5 (Song et al., 2018), but close to 9.4 – 10.7 (Dzepina et al., 2015), please rephrase.

We made a mistake here and the sentence has been revised as follows.

"Comparing with other studies, the DBE values in our filter are relative close to the results from biomass burning aerosol and aerosol samples from remote sites (Table 2)."

11. Line 225-226, "The AImod, ..., was correspondingly higher in F43 which contained 49.1% aliphatic (60.4% in F30), 45.9% olefinic (36.8% in F30), and 5.1% aromatic compounds (2.9% in F30)." What was correspondingly higher in F43? The total number of the following three compounds? Consideration the higher number in F30 for aliphatic, the current expression is ambiguous.

Agree. This sentence has been revised as follows.

"…was correspondingly higher in P2 as illustrated by its higher contribution of olefinic (75.0% vs. 73.9% for P2 and P1) and aromatic compounds (10.3% vs. 7.7% for P2 and P1) (Table 1)."

12. Line 237-279, a total of 4554 and 5192 molecular formulas was identified for F30 and F43 and existed 3700 common molecular formulas, however, the unique molecular formulas were just 619 and 1142 for the two filter rather than the rest 4554-3700 and 5192-3700, please modify or add specific values to Table 1.

We have made the number consistently in the updated manuscript.

13. Line 258, "A threshold DBE/C value of 0.7 usually serves as a criterion to identify species with condensed aromatic ring structures", please added references.

Done.

14. Line 269-270, the RI values mentioned here are in this study rather than in the reference, please declare.

Agree. The RI values are all listed in Table 3.

15. Line 313, there is no information about the 1N and 2N compounds in the Table 1.

We made a mistake here and revised accordingly.

16. Line 372, the 7.6% contribution of nitrogencontaining compounds to PM1 is from Zhang et al. (2018) for the entire long period rather than the two filter period (9% in Fig. 1), please added the reference to give a clear description.

Revised as the reviewer suggested.

"Due to the relative higher mass concentration and higher contribution of nitrogen-containing compounds (9% of $PM_1$) during these two periods based on HR-ToF-AMS results (Zhang et al., 2018), it is believed that aerosol transported to the Himalayas have important application in atmospheric radiative forcing."

Minor comments:
Line 18, change "are" to "were"
Line 22, change "significant" to "significantly"
Line 25, change "diagnose" to "diagnostic"
Line 26, change "highly" to "high"
Line 29, change "biomass mass burning" to "biomass burning"
Line 31, remove "are"
Line 53, change "could" to "can"
Line 57, remove "of", change "elevation" to "elevated"
Line 84, change "includes" to "included"
Line 92, change "with an average temperature..." to "and an average temperature..."
Line 101, 219, change "relative" to "relatively"
Line 218, change "than than of F30" to "than that for F30"
Line 260, change "were" to "have"
Line 366, change "particular" to "particulate"

All the minor issues above have been revised accordingly.

**Reviewer #2**

The authors of this paper reported sampled water-soluble organic matter (WSOM) at a high elevation site of Himalayas. They identified CHO and CHON compounds to be primary molecular compositions of WSOM. The paper provided important information on organic aerosols which could contribute to climate change and atmospheric oxidation over the Tibetan Plateau. The article should be published in ACPD. At this stage, I have only several minor comments as outlined below.

1. The authors attributed the sources of WSOM to biogenic volatile organic compounds and biomass mass burning. Given the lower temperature (~5.7∘C), BVOCs might not be readily formed and unlikely transported from distance sources as well. Or there is a high level of solar radiation/photosynthetically active radiation (PAR) in the Himalayas which favors the BVOC formation? Authors perhaps need to make a comment on this point.

In the manuscript, we emphasize that the oxidation products of biogenic volatile organic compounds (BVOCs) could be emitted and/or formed in low elevation regions in the south Asia. Oxidation products of BVOCs was found in inland of India (Fu et al., 2010) and in low elevation of Himalayas (Stone et al., 2012) during monsoon and post monsoon seasons. Aerosol in low elevation of south Asia could be transported to Himalayas and the inland of Tibetan Plateau through favorable large-scale atmospheric circulation (Zhang et al., 2017a) and regional/local meteorological conditions. Biogenic aerosol was also identified in aerosol collected in Namco Station which is far away from south Asia and has higher elevation than our sampling site (Ding et al., 2014). Actually, BVOCs were also observed in the mountain area of southern Himalayas at high elevation area (5050 m) (Ciccioli et al., 1993). In our results, products from ozonolysis of α-pinene were found with high relative intensity. Therefore, we believed that these signals were real, and these oxidation products could be transported to our sampling site. To clarify this point, we add a few sentences and references to support this conclusion (line 282 – 288).

2. I don't think that the measurement site of this study was located in the free troposphere (Table 2ïijˇ Nand perhaps the sampling sites in other studies listed in Table 2). The free troposphere means the troposphere above the boundary-layer. Although the elevation of the sampling site is 4275 m, it is on the ground surface and hence within the boundary layer.

Agree. The sampling site is in the boundary-layer which was frequently impacted by long-range transport air mass from low elevation regions. We change them to remote sites in Table 2.

Please check the language carefully.
There are quite a lot of grammar and spelling errors. For example,
Line 51 'is' should be 'are';
Line 53, 'could from';
Line 57, 'in the southern of';
Line 158 and 164, 'were' should be 'was',
Line 171, 'person correlation efficiency';
Line 181, 'transport' should be 'transported';
Line 698 (Fig. 2 captions), 'ground level of 1000m', you mean 1000 m above the ground level?

All the language issues have been revised and we have carefully checked the language throughout the manuscript.

**Reviewer #3**

The manuscript presented by Y. An et al. presents the detailed molecular chemistry of two samples collected from the Qomolangma Station in the Himalayas. The detailed molecular chemistry was derived from ultra-high resolution FT-ICR mass spectrometry measurements following electrospray ionization to generate positive ions. The authors discuss the molecular composition of the two samples and compare them carefully to previously published studies using similar approaches. Specifically, they found an increased degree of unsaturation of the prominent species in their study. As the authors suggest, these molecules may contribute to aerosol absorption. The manuscript is well prepared and the methodology is technically sound. However, I recommend the authors consider revisions to the manuscript to address method limitations pertaining to the ionization (potential artifacts, differences between + and -, anticipated functional groups, etc.) and discuss more specifically the significance of the results (both with respect to the implications and the limitations).

The limitation of our study only using ESI(+) are presented in the updated manuscript in method section and implication section. The response to potential artifacts and differences between ESI+ and ESI- is presented in specific question below. In the implication section, we also emphasize the significance of our results of the molecular compounds for the radiative forcing in the Himalayas and remove the content of biogeochemical effects.

Specific major comments without any predetermined order:
1.  It appears that many assumptions about the ionization method were made in the data interpretation. Those assumptions are not explicitly stated and may be incorrect. (i) For example, NH4+ is a common cation that readily adducts to molecular to assist in forming positive ions (similar to Na+). Please explicitly state your assumptions regarding this possible artifact. (ii) CHON compounds observed in ESI are expected to vary with the ionization mode. For example, reduced N (e.g., amino functional groups) are not expected to be observed in the negative ion mode. Likewise, oxidized N (e.g., nitrate functional groups) are not expected to be observed in the positive ion mode. Please explicitly state your assumptions regarding the ionization method and possible differences between ESI positive ions and ESI negative ions.

Thank you for your point this out. We state our assumptions regarding the potential compounds ionized in ESI positive mode in the updated manuscript (line 134-141). For the adduct of $NH_4^+$, we cannot exclude the possibility that some of compounds many form $[M + NH_4]^+$, however, the possibility of this formation was low by comparing with formation of $Na^+$ adducts because the binding strength of oxygen containing organic molecules with $Na^+$ ion is expected to be stronger than that of $NH_4^+$ ion. Highly oxygenated molecules that contain multiple peroxide functionalities were found to be readily cationized by the attachment of Na+ during electrospray ionization operated in the positive ion mode (Zhang et al., 2017b). In addition, the concentration of ammonium was low and we control the concentration of SPE effluent WSOM to be ~0.2 mg/mL which was not too concentrated for artifact adducts. The degree of ionization of nitrophenolic compounds at low acidic condition could be high and the pH of our mobile phase was between 2 and 3 which was thought to favor for nitro-phenolic compounds ionization.

2. Due to differences in the ESI ionization process (positive vs negative), the direct comparison of the data can be difficult. Please be sure to check the ionization mode of referenced datasets and discuss the method limitations associated with the datasets and the resulting limitations on the conclusions.

Agree. The ESI mode in each reference has been listed in Table 2. The direct comparison of molecular composition between different data in the manuscript was mainly based on the same ESI mode.

3. How were the molecular formulas and their homologous series formed from biogenic VOCs and biomass burning identified?

We remove the content of homologous series formed from biogenic VOCs and biomass burning emissions in the abstract and only emphasize the marker of these aerosol sources. The homologous series identified in KMD vs. KM plot were only focused on the potentially predominated compounds.

4. The phrase "important implications" (line 32) is an empty phrase. Please be more specific with the inferred implications and impacts associated with the studied molecular classes.

This sentence has been revised as follows.

"The high DBE and high fraction of nitrogen containing aerosol can potentially impact aerosol light absorption in this remote region."

5. What do your sample names indicate or represent? Consider changing the samples names to be more descriptive.

The names of samples are now denoted as P1 (period 1) and P2 (period 2), respectively, and consistent throughout the manuscript.

6. What is meant by "pristine region"?

This has been changed to "remote region".

7. The literature review describing the significance of light absorbing aerosol is severely out of date.

We have enriched this part and updated the references as follows.

"Brown carbon can originate from primary emission and/or secondary process, and have an increasing contribution (up to ~20%) to the light absorption in recent years (Laskin et al., 2015, and reference therein). Due to the light absorption of brown carbon is strongly depended on their molecular structure, light absorbing compounds at molecular level were explored during recent years and found that nitrogen-containing compositions are important brown carbon compounds (Lin et al., 2016; Lin et al., 2017)."

8. The phrase "Many studies" requires more than just one example reference.

We added one more literature here.

9. The goal of the paper is what exactly?

We add one sentence to mention the purpose of this study as follows.

"In this study, we focus on the molecule composition of water soluble organic compound in fine particle aerosol in the Himalayas using positive mode ESI-FTICR MS and evaluate the sources, chemical processing, and potential impact of aerosol in this region."

10. What is the relevance of discussing the tourist season?

This sentence has been removed.

11. Ultrasonic baths can introduce reactive oxygen species. What care was taken to avoid extraction artifacts?

We added ice during ultrasonic extraction and kept the sample immersing in the mixture of ice and water. This information is updated in the manuscript as follows.

"The sample tubes were immersed in the mixture of ice-water during ultrasonic extraction to prevent potential chemical reaction."

12. Please discuss the SPE recovery.

We did not measure the SPE recovery for this study. Based on the previous studies, water soluble organic carbon recoveries ranged between 20% and 65% for the different SPE sorbents (Dittmar et al., 2008; Green et al., 2014; Raeke et al., 2016). PPL usually has higher recovery than C18 or HLB for subsequent FT-ICR MS analysis (Green et al., 2014). In addition, the FT-ICR mass spectra of the original sample and the SPE extracts did not differ significantly in their molecular weight distribution, but they showed sorbent specific differences in the degree of oxygenation and saturation; The selective enrichment of freshwater WSOM by SPE is less critical for subsequent FT-ICR MS analysis, because those fractions that are not sufficiently recovered have comparatively small effects on the mass spectra (Raeke et al., 2016). We add one sentence to support the usage of PPL cartridge in the updated manuscript as follows.

"PPL cartridge generally has the best properties for WSOM enrichment for subsequent FT-ICR MS analysis (Raeke et al., 2016)."

13. Please discuss the steps that were taken to avoid ESI artifacts?

The possible artifacts of positive ESI method is the formation of adducts such as sodium, ammonium and so on which could complex mass spectrum. These artifacts could try to be avoided by removing the inorganic salt by SPE before measurement. In our study, we apply SPE concentration to eliminate the inorganic salts as more as possible, although we cannot exclude the possibility that some of compounds may form from these artifacts. In addition, we control the concentration of SPE effluent WSOM to be ~0.2 mg/mL which was not too concentrated for artifact adducts. We revised the sentence in the updated manuscript to mention this as follows.

"Prior to FTICR MS analysis, the extraction was concentrated and purification using PPL (Agilent Bond Elut-PPL cartridges, 500 mg, 6 mL) solid phase extraction (SPE) cartridges for water soluble organic matter (WSOM) to avoid possible ESI artifacts. PPL cartridge generally has the best properties for WSOM enrichment for subsequent FTICR MS analysis (Raeke et al., 2016). In addition, we control the concentration of SPE effluent to be ~0.2 mg/mL which was not too concentrated for artifact adducts."

14. The parameters associated with your "custom software" need to be more thoroughly described. How did you eliminate ambiguous formula assignments?

The formula assignment is controlled by the mass accuracy up to ±1.5 ppm as well as a number of criteria including the isotope pattern and elemental ratios, such as H/C, O/C, N/C, S/C, and DBE/C in the ranges of 0.3-3.0, 0-3, 0-0.5, 0-0.2, and 0-1. We add this information in the updated manuscript as follows.

"The ions detected in filter blank were subtracted and molecular formulas in the samples were assigned to all ions with signal-to-noise ratios of greater than 10 with a mass tolerance of ±1.5 ppm using custom software. Molecular formulas with their maximum numbers of atoms were defined as: 30 $^{12}$C, 60 $^{1}$H, 20 $^{16}$O, 3 $^{14}$N, 1 $^{32}$S, 1 $^{13}$C, 1 $^{18}$O and 1 $^{34}$S. Identified formulas with H/C, O/C, N/C, S/C, and DBE/C ranged in $0.3 - 3.0$, $0 - 3$, $0 - 0.5$, $0 - 0.2$, and $0 - 1$, were selected, and formulas containing isotopomers (i.e., $^{13}$C, $^{18}$O or $^{34}$S) were not considered."

15. What is meant by the "processing error" mentioned in line 161? How did you ensure that the error did not affect the molecular composition?

We are sorry for your confusion by this sentence. Actually, we made a mistake during the weighting of the samples which was not related with chemical measurement. The processing error was that we did not balance the filters at the same conditions as that before sampling. So that the gravimetrically data of the filters were not correct.

16. What is the procedure for identifying the NOA compounds with HR-ToF-AMS?

The NOA compounds identified in HR-ToF-AMS is based on positive matrix factorization (PMF) analysis which was included in details in our previous paper (Zhang et al., 2018). Basically, the identification of NOA factor was based on its special mass spectrum, diurnal pattern, and correlation with other tracers. We mention this information in the updated manuscript as follows.

"The OA was comprised by biomass burning emitted OA (BBOA), nitrogen-contained OA (NOA), and more-oxidized oxygenated OA (MO-OOA) decomposed by positive matrix factorization (PMF) analysis (Fig. 1). The details on PMF analysis can be found in Zhang et al. (2018)."

17. How was the influence of potential fresh OA inferred?

The fresh OA here mean less oxidized OA which was inferred based on the short trajectory distance and higher contribution of BBOA in P2 than P1. In order to clarify this, we revise this sentence as follows.

"The air mass during P2 was partly (13%) transported with low wind speed and short distance (less than 100 km) which could contain some fresh OA as illustrated with higher fraction of BBOA."

18. In the discussion of common ions, the authors assume that the two samples have a similar aerosol source?  What if instead, the common compounds are simply not marker compounds. Complex mixtures are expected to have many ions in common.

We agree that complex organic mixture has many ions in common. The two samples in this study collected during one long-range transport event at different stages. Although, the chemical characteristics between these two samples were kind of different, the aerosol sources for them could be similar based on trajectory analysis and AMS-PMF results. In addition, the mass spectra and ion composition from FTICR MS between them were also very similar. Therefore, for more confident on our analysis, we extract the common formula for further discussion.

19. What is the balance of source contribution vs. aerosol aging in these samples.

It is hard to know the exactly ratio between source contribution and aerosol aging. We analyzed the potential aging processes including photo reaction, aqueous reaction and dark reaction during the transport in Zhang et al., (2018), and found these reactions could be important for aerosol aging.

20. I'm surprised that the long-range transported aerosol reported in Dzepina et al. is similar to the samples reported here.  How is that observation justified with respect to the transport pathways?

We agree that this sentence is confused and therefore delete it in the updated manuscript.

21. The "distinct group of CHON aromatic compounds" in the lower left of the VK diagram may be incorrect assignments. What are the limits for the DBE range?

We set the limit of DBE/C of 0-1 and this group CHON is in this range. Consider the extremely high carbon number (>40), we remove this group.

22. What is the significance of the difference in the max abundance between H+ and Na+ type ions?

The $[M + Na]^+$ compounds could be carboxylic acid groups that readily form $[M + Na]^+$ ions in the positive electrospray ionization mode. Since we delete the high abundance group of $[M + Na]^+$, the sentence related with $[M + Na]^+$ has been removed.

23. The detailed description given over lines 279-298 is quite tedious. Perhaps some of these formulas can be better presented in a figure or table?

A new table (Table 3) is added in the updated manuscript to present all the formula mentioned in the text.

24. What is the significance of 1N vs 2N?

We compare the compounds of 1N and 2N to get the information of structure and chemical formation of nitrogen-containing compounds. As shown in the manuscript, the elemental ratios (O/C, H/C, O/N, and DBE) were different for 1N and 2N compounds, and the potential formation for 1N and 2N compounds are discussed in the manuscript.

25. How do you observe acidic N in the positive ion mode?

We remove the citation of amino acids here. The CHON compounds observed in our positive ESI mode could contain reduced N functional groups (e.g., amines), which are preferentially ionized in ESI+ mode. Similar results which observed CHON compounds in positive ESI mode was also found in previous study in biomass burning influenced aerosol (Lin et al., 2012; Wang et al., 2017). In addition be nitro-phenolic compounds are also likely ionized in ESI+ mode in the acidic mobile phase. We have made the CHON compounds more clear in the updated manuscript.

26. The statement in lines 330-333 is not convincing. Please rephrase and add more evidence or description.

Agree. We have added a few sentences here as follows.

"Lin et al. (2017) found aged biomass burning aerosol in the present urban oxidants (such as $NO_x$) could result in higher fraction of CHON compounds comparing to the fresh biomass burning aerosol. Considering the high influence of biomass burning emission in the Himalayas (Zhang et al., 2018b), the CHON compounds in our study were probably related with biomass burning emissions. Recent studies have proven that burning of mixed biomass fuels in Nepal could emit amount of nitrogen species such as $NH_3$, $NO_x$, HCN, benzene, and organics, and the emission factors for these species are higher than that of wood (Stockwell et al., 2016; Jayarathne et al., 2018). In addition, it is likely that smoldering burning of bio-fuels in high elevation area is also responsible for the presence of many nitrogen-containing compounds in BBOA (Chen et al., 2010)."

27. Lines 351- 353: How does the sample matrix effect the observation of ions in ESI?

This explanation was deleted.

28. Where are the major products of BVOC as mentioned in lines 359-363?

These formulas were monoterpene products with $NO_3$ radical and these information is listed in Table 3.

29. The discussion of the research implications can be enhanced with a deeper discussion of the molecular composition and method limitations. What other observations of absorbing species have been made in the Himalayas?

So far there is not molecular based light absorption measurement. We have improved the discussion on the implications which include the molecular composition and method limitations as follows.

"More comprehensive methods are needed in the future for identifying BrC in the Himalayas due to the chemical complexity of BrC. For example, the BrC extraction is highly dependent on the used solvent and water insoluble OA can contribute higher light absorption than water soluble OA (Chen and Bond, 2010). In addition, Budisulistiorini et al. (2017) found that a number of compounds can dominate the light absorption of BrC, although they have a minor contribution to the aerosol mass. Therefore, it is important to know the exactly chromophores of BrC which can be obtained by combining with high performance liquid chromatography, light absorption measurement with a photodiode (PDA) detector, and chemical composition with high resolution mass spectrometer (HPLC-PDA-HRMS system) (Lin et al., 2016). For mass spectrometry analysis, different ionization sources are also favorable for different compounds, such as ESI only detect a part of polar compounds; non-polar compounds which could dominated the contributing of BrC, can be measured using atmospheric pressure photo ionization (APPI) (Lin et al., 2018). Recent study indicate that over 40% of the solvent-extractable BrC light absorption is attributed to water insoluble, non-polar to semi-polar compounds such as PAHs and their derivative (Lin et al., 2018). In contrast, the polar, water-soluble BrC compounds, which are detected in ESI, account for less than 30% of light absorption by BrC (Lin et al., 2018)."

30. The implications regarding nutrients and biogeochemical cycling are beyond the scope of the current research and seem to be a bit too ambitious. Please revise.

Agree. This information has been removed.

31. Related to the previous comment, did you study deposition?

Yes. We collected precipitation samples during this field study and the project we involved cover several direction including biogeochemical cycling. But we agree that the biogeochemical cycling is out of the range of this study and remove this information in the updated manuscript.

32. Again, what type of N did you study with your analytical method. Please be clear with the limitations and assumptions that are necessary.

The nitrogen compounds in our study were likely reduced nitrogen and nitro-aromatic compounds. To make more clarify this point, a few sentences are added in 3.3.2 section.

33. How are the measurement sites defined? The listed free troposphere sites do not consistently sample free tropospheric air. In each case, seasonal factors may play a strong role in the height of the boundary layer.

Agree. Free troposphere sites should above boundary layer height and have less influence from it. Our site and some other sites in Table2 were strongly influenced by the air from boundary layer. We have revised these sites in Table 2 to remote sites.

34. Figure 3 appears to contain several high intensity regularly spaced peaks that are not associated with the sample. Please remove or flag these peaks as contaminants.

This high intensity group has been removed.

35. Please add the specific details regarding your treatment of blank samples to the methods section.

Agree. The description for the treatment of blank sample is added in the method section.

"One procedure blank was also adopted in this study as like that of aerosol samples to subtract the potential background."

"The ions detected in filter blank were subtracted and molecular formulas in the samples were assigned to all ions with signal-to-noise ratios of greater than 10 with a mass tolerance of ±1.5 ppm using custom software."

36. How are the FT-ICR MS ions related to the fragment ions from HR-ToF-AMS (as shown in figure 4)?

The connection between FT-ICR MS ions and HR-ToF-AMS frags is through the index of carbon oxidation state (OSc) (Kroll et al., 2011). The shaded ovals indicate locations of different ambient organic aerosol classes as determined from factor analysis of HR-ToF-AMS data, which the nC is estimated from volatility measurements (Kroll et al., 2011).

Additional minor comments:
Line 19: DBE = double bond equivalents; DBE is plural not singular

Revised as suggested by reviewer.

Line 31: "high nitrogen containing of aerosol" is unclear. Please rephrase.

Revised to "high fraction of nitrogen containing of aerosol".

Line 32: "important implications" (use plural)
This sentence has been changed accordingly.

 "The high DBE and high fraction of nitrogen containing aerosol can potentially impact aerosol light absorption in this remote region."

Line 32: "and the biogeochemical cycle" (insert article)

The biogeochemical cycle content has been removed.

Line 42: "Accompany" is awkward. Please rephrase.

Revised to "Under favorable atmospheric circulation".

Line 43: "to the Himalayas" (insert article)

Done and check throughout the manuscript.

Line 46: "essential" is awkward. Please rephrase

Change to negative.

Line 172: What is the IA method?

For AMS study, there is an improved method to calculate the elemental ratios in recent year. The full name of IA method and reference are shown in the manuscript.

Line 181: "transported" (verb tense)

Done.

Line 185: Fix typo.

Done.

Lines 247 & 249: "molecular weight" not "molecule weight"

Revised and throughout the manuscript.

Lines 256 & 258: Typo? Did you mean to use Cw or C in these two sentences?

Change to C.

Line 261: "carbon oxidation state" (lower case "C")

Revised as suggested by the reviewer.

Line 307: "average O atoms contained in" (plural and verb tense)

Revised as suggested by the reviewer.

Line 368: "two compounds" or "two compound groups"

Change to two compound groups.

Line 373: "believed" (verb tense)

Done.

Table 1 (and Figure 3): Are the values shown for F30 and F43 for all ions or only unique ions?

The values shown in Table 1 and Figure 3 are for all ions. The captions of Table 1 and Figure 3 have been improved.

Figure 1: What is the purpose of the blue shading behind the pie chart?

The blue shading has been removed.

Figure 1 (and elsewhere): Please remember to define all of the acronyms used in the figure within the figure caption.

Agree. All the acronyms used in the figures have been shown with the full name.

Figure 2: Please include the vertical profile for the back trajectories

The vertical profile has been colored on each cluster using air pressure data.

[revised manuscript text omitted]